# Ice flood disaster risk assessment of the Inner Mongolia reach in the Yellow River based on unascertained measure theory-variable fuzzy sets

Yu Deng[1,2,3]*, Lu Jiang[1], Juan Wang[1]

1 School of Water Resources and Transportation, Zhengzhou University, Zhengzhou, Henan, China,
2 Yellow River Institute of Hydraulic Research, Zhengzhou, Henan, China, 3 Technological Innovation Center for Levee Safety and Disaster Prevention, MWR, Zhengzhou, Henan, China

* dengyu@vip.sina.cn

**Editor:** Beata Calka, Military University of Technology Faculty of Civil Engineering and Geodesy: Wojskowa Akademia Techniczna im Jaroslawa Dabrowskiego Wydzial Inzynierii Ladowej i Geodezji, POLAND

## Abstract

Ice flood disasters represent the most severe natural hazards affecting the Yellow River during winter-spring periods, posing significant threats to residents' safety and impeding regional socioeconomic development in the basin. Despite their critical implications, systematic risk assessment methodologies tailored for Yellow River ice floods remain underdeveloped. Current approaches inadequately address the inherent uncertainty and fuzziness characteristics of ice flood disaster systems, thereby compromising assessment accuracy. To bridge this gap, this study develops an integrated risk assessment model combining Unascertained Measure Theory (UMT) and Variable Fuzzy Set Theory (VFST). The hybrid model was implemented in three typical reaches of the Yellow River's Inner Mongolia section: Tokto County, Jungar Banner, and Qingshuihe County. Risk evaluation results classified Tokto County and Qingshuihe County as "medium-risk" areas, while Jungar Banner was identified as a "high-risk" zone. Validation analyses confirmed strong consistency between model outputs and historical disaster patterns, demonstrating the framework's reliability. This research establishes a novel methodological foundation for ice flood risk quantification and provides actionable insights for disaster mitigation strategies in cold-region river systems. The proposed approach offers technical support for enhancing flood management decision-making processes in the Yellow River Basin.

## Introduction

Ice flood disasters predominantly occur during winter and spring seasons, initiated by thermal fluctuations that trigger ice jam formation. These jams obstruct river channels, inducing abrupt upstream water level surges and subsequent flooding—a phenomenon prevalent in high-latitude river systems. In China's cold climatic zones north of 30°N, approximately 70% of rivers undergo winter-spring freeze-up cycles,

**Data availability statement:** All relevant data are within the paper and its Supporting information files.

**Funding:** This work was supported by the National Natural Science Foundation of China under grant U23A2012 and 52509096, the Yellow River Water Conservancy Research Institute basic research funds under grant HKY-JBYW-2022-08.

**Competing interests:** NO authors have competing interests.

with the Yellow River Basin's Inner Mongolia reach experiencing particularly severe ice flood events.

Early research concerning Yellow River ice jams could be traced back to the 1990s. Shen et al. [1] developed the RICE model, a numerical framework integrating one-dimensional steady-state flow dynamics, heat transfer mechanisms, and ice transport equations. Subsequent enhancements introduced unsteady flow dynamics and ice regime parameters, resulting in the advanced RICEN model for ice regime simulations in China's Yellow River Basin [2].

In parallel work, Shen et al. [3] conducted experimental investigations using low-density particulates to analyze particle morphology effects on ice transport capacity. Their methodology combined Eulerian finite element analysis with smoothed particle hydrodynamics (SPH) to develop 2D numerical models simulating surface ice dynamics and fluvial blockage formation [4].Chen et al. [5] proposed a fuzzy-optimized neural network incorporating hydrothermal, hydrodynamic, and geomorphological parameters to predict ice phenology (freeze-up/break-up timing) in the Yellow River's Inner Mongolia section, demonstrating robust validation results.

Lin [6] implemented fixed-wing unmanned aerial vehicle (UAV) systems for experimental ice monitoring, pioneering UAV applications in Yellow River ice surveillance. Through micromechanical modeling and macro-mechanical property simulations, Deng [7] established and validated the representative volume unit (RVE) methodology for quantifying Yellow River ice strength. Building on physical testing, Deng [8] further created a numerical model for three-point bending fracture analysis, elucidating fracture toughness characteristics of river ice.

Advancing understanding of Yellow River ice phenomena has revealed two systemic challenges: inherent uncertainty in flood dynamics and complexity in environmental interactions. Recent climate variability and anthropogenic disturbances have exacerbated thermal instability, amplifying ice flood unpredictability. Concurrently, diversified river morphology and expanding hydraulic infrastructure have compounded system uncertainties. These synergistic factors intensify challenges for riverside disaster prevention and mitigation efforts.

Risk assessment emerges as a critical tool for ice flood management, enabling scientific quantification of spatial risk levels along river reaches. Strategic implementation of targeted prevention measures and optimized resource allocation, guided by assessment outcomes, significantly enhances flood resilience. Consequently, advancing risk assessment methodologies for Yellow River ice floods holds paramount importance. Wu et al. [9] created a hybrid risk model combining projection pursuit, fuzzy clustering, and accelerated genetic algorithms, applied to 1991–2010 ice flood data in the Ningxia-Inner Mongolia reach. Luo [10] employed a dynamic multiindex GM (1, 1) model to forecast ice jam/dam risks during 2013–2015 in the same region. Wang [11] developed a random forest-based assessment framework for systematic risk evaluation in Inner Mongolia's flood-prone zones. Hu [12] addressed composite value inflation limitations in catastrophe theory methods through model modifications, subsequently applying enhanced techniques to Inner Mongolia ice flood risk analysis.

Contemporary flood risk assessment research has achieved substantial theoretical advancements across multiple methodological frameworks. Nevertheless, current approaches insufficiently resolve the intrinsic uncertainty and fuzziness inherent to ice flood disaster systems, requiring improved evaluation accuracy. UMT provides an effective mechanism for processing incomplete objective information under data-deficient scenarios. This methodology enhances assessment precision through minimized subjective bias, with demonstrated cross-disciplinary applicability in risk quantification studies [13,14]. Differing fundamentally from classical fuzzy set theory, VFST introduces adaptive coefficients enabling dynamic membership function adjustments. This innovation facilitates enhanced characterization of transient fuzzy system behaviors. The approach has proven particularly advantageous in hydraulic engineering evaluations and water resource risk management applications [15,16].

Drawing upon the aforementioned analysis, this paper combines the UMT with the VFST to conduct in-depth research into the flood risk in the Inner Mongolia section of the Yellow River in Tokto County, Jungar Banner, Qingshuihe County, which are prone to flood danger, and presents a new methodology and perspective for evaluating flood disaster risk in the Inner Mongolia section of the Yellow River by establishing a flood hazard risk assessment model and carrying out a systematic assessment.

This study employs the following organizational framework to ensure technical clarity and reader accessibility. Section 2 establishes the conceptual architecture and systematic assessment index system, defining critical parameters for ice flood risk quantification. Section 3 delineates the methodological framework, incorporating theoretical integration of UMT with VFTS principles and operational development of the hybrid evaluation model. Section 4 implements the proposed methodology across three strategically selected regions along the Yellow River's Inner Mongolia reach (40°15'N–40°45'N, 111°02'E–111°46'E), executing computational analyses to identify critical spatial risk patterns and validate model efficacy. Section 5 synthesizes principal scientific contributions and practical implications derived through systematic investigation, while proposing adaptive flood management strategies for cold-region river systems

## Procedures

The specific steps of this study are as follows:

- Gather relevant hydrological, meteorological, socio-economic, and engineering data pertaining to the study area.

- Establish the index system and classification criteria for evaluating ice flood disaster risk.

- Ascertain the weight of each index via the Analytic Hierarchy Process (AHP).

- Create a coupled evaluation model grounded in UMT and VFST.

- Compute the risk levels of each evaluation object through the developed risk assessment model.

Ice flood disaster risk assessment constitutes a complex analytical process requiring systematic index framework development as its foundation. The selection of evaluation parameters must follow established criteria including scientific validity, hierarchical organization, systemic coherence, and practical applicability. Drawing upon regional disaster system theory [17] and incorporating causative mechanisms of ice flood disasters alongside existing assessment frameworks [18], this study establishes a three-tiered evaluation system (Fig 1). The primary tier (objective layer) focuses on ice flood disaster risk quantification. The secondary tier (criterion layer) comprises four components: disaster-inducing factors, disaster-breeding environment, hazards-bearing, and disaster response capability. Disaster-inducing factors and predisposing conditions collectively characterize hazard magnitude, while hazards-bearing bodies quantify regional vulnerability. Disaster response capability evaluates infrastructural resilience. The tertiary tier (index layer) contains 13 operationalized parameters across these criteria. Each evaluation metric was systematically categorized into distinct classification tiers using longitudinal extremum analysis (encompassing maximum, minimum, and mean values) derived from multi-decadal datasets, with operational thresholds explicitly defined in Table 1.

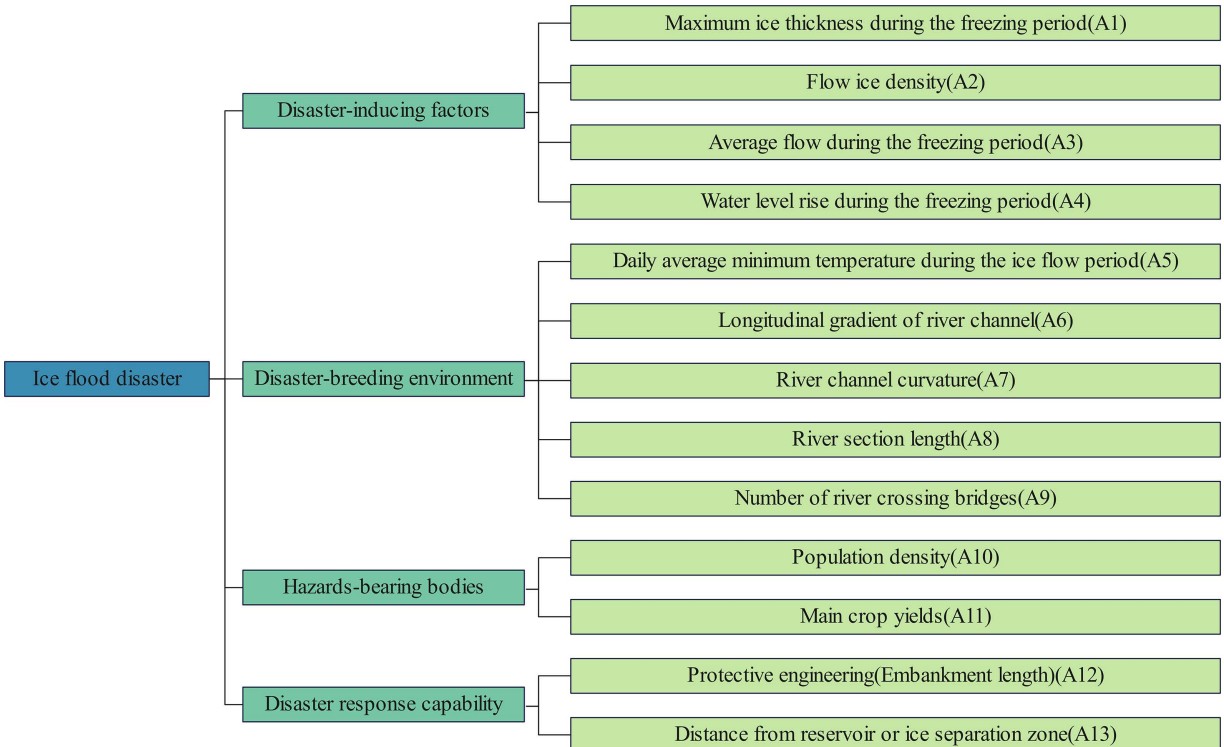

**Fig 1. Risk assessment index system for ice flood disasters.** The figure illustrates the hierarchical framework of the ice flood disaster risk assessment index system developed for the Inner Mongolia reach of the Yellow River, together with its specific indicators at each level.

## Method

### Analytic hierarchy process

In this research, the weights of the indexes are determined using the AHP. Owing to its high degree of comprehensiveness and wide applicability, AHP is widely regarded as a reliable method for assigning weights. The detailed procedures are as follows:

- Based on the established evaluation index system, pairwise comparisons of the indexes are performed within each level to form the judgment matrix;

- Solve the judgment matrix in order to calculate the maximum eigenvalue and eigenvector, and carry out consistency testing;

- Aggregate the weights of each level to acquire the overall weights of the lower-level indexes.

### The entropy weight method

The entropy method is an approach to assigning weights based on the impact of each indicator's value variation on the overall system. In information theory, entropy measures uncertainty, and the greater the entropy value of an indicator, the smaller its weight. When addressing the problem of multi-indicator weighting, the entropy method helps eliminate biases from subjective assignments, improving the objectivity and accuracy of the evaluation results. The specific steps are as follows: standardize the data according to the indicator type (benefit-oriented or cost-oriented); process the standardized data to calculate the entropy value for each indicator; and determine the weight of each indicator based on its entropy value.

**Table 1. Classification standard of risk assessment index of ice flood.**

| Index | Risk level | | | | |
|---|---|---|---|---|---|
| | Very low risk | Low risk | Medium risk | High risk | Very high risk |
| Maximum ice thickness during the freezing period(m) | <0.2 | 0.2-0.4 | 0.4-0.6 | 0.6-0.8 | >0.8 |
| Flow ice density(%) | <30 | 30-50 | 50-70 | 70-90 | >90 |
| Average flow the during freezing period(m³/s) | >900 | 700-900 | 500-700 | 300-500 | <300 |
| Water level rise during the freezing period(m) | <0.5 | 0.5-1 | 1-1.5 | 1.5-2 | >2 |
| Daily average minimum temperature during the ice flow period(°C) | >-14 | (−14,-16] | (−16,-18] | (−18,-20] | <−20 |
| Longitudinal gradient of river channel(‰) | >0.4 | 0.3-0.4 | 0.2-0.3 | 0.1-0.2 | <0.1 |
| River channel curvature | <1.2 | 1.2-1.4 | 1.4-1.6 | 1.6-1.8 | >1.8 |
| River section length(m) | <50 000 | 50 000-100 000 | 100 000-150 000 | 150 000-200 000 | >200 000 |
| Number of river crossing bridges | <1 | 1-3 | 3-5 | 5-7 | >7 |
| Population density(person/km²⁾ | <50 | 50-200 | 200-400 | 400-600 | >600 |
| Main crop yields(kg/m²) | <0.45 | 0.45-0.5 | 0.5-0.55 | 0.55-0.6 | >0.6 |
| Protective engineering(Embankment length)(m) | $<3\times10^4$ | $3\times10^4$-$6\times10^4$ | $6\times10^4$-$9\times10^4$ | $9\times10^4$-$1.2\times10^5$ | $>1.2\times10^5$ |
| Distance from reservoir or ice separation zone(m) | $<10^5$ | $10^5$-$3\times10^5$ | $3\times10^5$-$5\times10^5$ | $5\times10^5$-$7\times10^5$ | $>7\times10^5$ |

## Uncertainty measurement theory

The ice flood disaster risk assessment framework incorporates 13 evaluation indexes. Let the evaluation object set $M=\{M_1, M_2, …, M_{13}\}$, where $M_m$ represents the m-th evaluation index. Each measurement value of the evaluation index is denoted by $x_m$. The measurement values are classified into five evaluation levels, represented as $C_1, C_2, …, C_5$, with the condition that $C_1 < C_2 < C_3 < C_4 < C_5$. Let $u_{mk} = u(x_m \in C_k)$ represent the degree to which the measurement value $x_m$ belongs to the evaluation level $C_k$. The uncertainty measure u satisfies conditions (1) – (3) [18]:

$$0 \leq u(x_m \in C_k) \leq 1 \tag{2}$$

$$u(x_m \in U) = 1 \tag{3}$$

$$u(x_m \in \bigcup_{l=1}^{k} C_1) = \sum_{l=1}^{k} u(x_m \in C_1) \tag{4}$$

The measurement matrix for a single index in ice flood disaster risk assessment, denoted as $u_{mk}$, can be expressed as follows:

$$\mathbf{u_{mk}} = \begin{pmatrix} u_{11} & \cdots & u_{1k} \\ \vdots & \vdots & \vdots \\ u_{m1} & \cdots & u_{mk} \end{pmatrix} \tag{5}$$

The comprehensive evaluation vector is then computed through:

$$\mathbf{u} = \mathbf{W u_{mk}} \tag{6}$$

Where: *W* is the index weight vector previously determined through entropy weighting.

## Variable fuzzy set theory

Chen proposed the VFST considering the relative nature and dynamic changes of fuzzy concepts, which provides a new method for addressing dynamic fuzzy problems [19]. The theory is based on two key concepts: the relative membership degree and the relative difference degree:

$$\overline{D}_{\tilde{A}}(i) = \mu_{\tilde{A}}(i) - \mu_{\tilde{A}_c}(i) \tag{7}$$

Where: *i* represents any element in the discourse domain *I*, $i \in I$; *I* is the discourse domain, which is the set of all possible elements under consideration in this evaluation; $\tilde{A}$ and $\tilde{A}_c$ represent attractive and repulsive properties respectively; $\mu_{\tilde{A}}(i)$ and $\mu_{\tilde{A}_c}(i)$ represent the relative membership degrees of attractive and repulsive properties respectively, given that $\mu_{\tilde{A}}(i) + \mu_{\tilde{A}_c}(i) = 1$; $D_{\tilde{A}}(i)$ is the relative difference between *i* and $\tilde{A}$.

Therefore, the relative membership degree of *i* to $\tilde{A}$ can be articulated as:

$$\mu_{\tilde{A}}(i) = (1 + D_{\tilde{A}}(i))/2 \tag{8}$$

The comprehensive relative membership degree of evaluation index *i* to level *k* can be calculated by the following formula:

$$\nu_k = F(\omega_m, \mu_{mk}) = \left\{ 1 + \left[ \frac{\sum_{m=1}^{13} [\omega_m(1 - \mu_{mk})]^p}{\sum_{m=1}^{13} (\omega_m \mu_{mk})^p} \right]^{\frac{\alpha}{p}} \right\}^{-1} \tag{9}$$

## UMT-VFST evaluation method

Although VFST effectively reflects the dynamic fuzziness of phenomena, the calculation of relative membership degrees necessitates initially ascertaining relative difference degrees, which frequently depend excessively on practical experience. The uncertainty measure in uncertainty measurement theory and the relative membership degree in VFST share analogous meanings to a certain extent: the closer the value is to 1, the higher the degree of consistency between the assessed entity and the specified evaluation tier. By substituting the uncertainty measure vector for the relative membership degree in the VFST, the two methods can be organically integrated, exploiting their complementary strengths to further enhance the scientific rigor and accuracy of the assessment.

The linear uncertainty measurement function is selected to construct the single index measurement matrix as follows:

$$\begin{cases} u_{mk} = \begin{cases} -x_m/(a_{k+1} - a_k) + a_{k+1}/(a_{k+1} - a_k) & a_k < x_m \leq a_{k+1} \\ 0 & x_m \geq a_{k+1} \end{cases} \\ u_{m(k+1)} = \begin{cases} 0 & x_m \leq a_k \\ x_m/(a_{k+1} - a_k) - a_k/(a_{k+1} - a_k) & a_k < x_m \leq a_{k+1} \end{cases} \end{cases} \tag{10}$$

Where: $x_m$ denotes the measured value of evaluation index $M_m$; $a_k$ and $a_{k+1}$ represent the *k*-th and *k*+1-th critical thresholds on the unascertained measure function.

Subsequently, the unascertained measure $u_{mk}$ is used to replace the relative membership degree $\mu_{mk}$, serving as a bridge to couple the unascertained measure theory with the variable fuzzy set theory.

$$\mu_{mk} = u_{mk} \tag{11}$$

 

Subsequently, by utilizing the variable parameter combinations in the VFST, the comprehensive membership degree under each parameter combination is ascertained, dynamically illustrating the risk levels of the evaluation regions.

$$\nu_k = F\left(\omega_m, \mu_{mk}\right) = \left\{1 + \left[\frac{\sum\limits_{m=1}^{13}\left[\omega_m(1-u_{mk})\right]^p}{\sum\limits_{m=1}^{13}\left(\omega_m u_{mk}\right)^p}\right]^{\frac{\alpha}{p}}\right\}^{-1}$$

(12)

Where: $\omega_m$ is the weight of the m index; $\alpha$ serves as the parameter for the optimization criterion, $\alpha$ is either 1 or 2; $p$ represents the parameter for distance measurement, $p$ is either 1 or 2. There are usually four combinations: ① $\alpha = 1$, $p = 1$; ② $\alpha = 1$, $p = 2$; ③ $\alpha = 2$, $p = 1$; ④ $\alpha = 2$, $p = 2$.

Considering that there are currently no clear standards or guidelines for the classification of flood disaster risk levels, this paper refers to the classification method used in flood disaster risk assessment [20]. The ice flood disaster risk is categorized into five levels, and the classification established utilizing level eigenvalues. Initially, the comprehensive membership degree obtained from Equation (10) is normalized as follows:

$$V_k = \nu_k / \sum_{k=1}^{5} \nu_k$$

(13)

The level eigenvalues are calculated based on Equation (12) as follows:

$$H = \sum_{k=1}^{5} \left(V_k k\right)$$

(14)

The level eigenvalues corresponding to the risk levels are shown in Table 2.

## Results and discussion

### Weight calculation

Following the established methodological framework, pairwise comparison matrices were constructed for hierarchical indexes. Through structured expert elicitation using Saaty's 9-point scale, relative importance ratings were systematically obtained. The parametric indices A1-A13 exhibit systematic correspondence with the constituent elements within the evaluation framework schematically illustrated in Fig 1. Validation analysis demonstrates that all judgment matrices satisfy consistency ratio (CR) thresholds (CR < 0.1), confirming the statistical validity and methodological robustness of the derived weight allocation. By combining the Risk assessment index system for ice flood disasters in Fig 1 and the index data in Table 3, the entropy weight method is used to calculate the objective weights of each indicator. The

Table 2. The corresponding relationship between the level eigenvalue H and the evaluation level.

| Evaluation level | Level eigenvalue H |
| --- | --- |
| Very low risk | 1-1.5 |
| Low risk | 1.5-2.5 |
| Medium risk | 2.5-3.5 |
| High risk | 3.5-4.5 |
| Very high risk | 4.5-5 |

**Table 3. Index value of ice flood disaster risk assessment in each region.**

| Index | Tokto County | Jungar Banner | Qingshuihe county |
|---|---|---|---|
| Maximum ice thickness during the freezing period(m) | 0.6 | 0.95 | 0.85 |
| Flow ice density(%) | 40 | 60 | 100 |
| Average flow the during freezing period(m³/s) | 460 | 455 | 425 |
| Water level rise during the freezing period(m) | 1.4 | 6.6 | 6.1 |
| Daily average minimum temperature during the ice flow period(°C) | −18 | −18.4 | −16.8 |
| Longitudinal gradient of river channel(‰) | 0.24 | 0.66 | 0.38 |
| River channel curvature | 1.06 | 1.92 | 1.26 |
| River section length(m) | 37 500 | 238 000 | 65 000 |
| Number of river crossing bridges | 4 | 13 | 3 |
| Population density(person/km²) | 152 | 44 | 32 |
| Main crop yields(kg/m²) | 0.511 | 0.49 | 0.335 |
| Protective engineering(Embankment length)(m) | 78 200 | 60 646 | 30 000 |
| Distance from reservoir or ice separation zone(m) | 97 000 | 73 000 | 45 000 |

comprehensive weight is then obtained by calculating the weighted average of the results from the AHP and entropy weight method, with equal weights (50% each) assigned to both methods. The finalized weight distribution pattern is visually presented in Fig 2.

### Risk grade calculation

This investigation focuses on the 2016 ice flood event at the reservoir tail reach of Wanjiazhaias a representative case study. The methodologically triangulated dataset integrates: (a) scholarlyreferences [12], (b) municipal statistical yearbooks, (c) multi-sensor measurements frommeteorological observatories, and (d) instrumented hydrological monitoring records, assystematically cataloged in Table 3.

(1) Single-index measurement function: First, calculate the measurement function for each index, that is, the single-index measurement function. By taking the maximum ice thickness as an example, the uncertainty measurement function of which is shown in Fig 3. Based on Equation 5, yields the single-index measurement for the maximum ice thickness during the freezing period in Jungar Banner as (0,0,0,0,1). Similarly, the single-index measurement matrix for the 13 ice flood disaster risk evaluation indexes in Jungar Banner is presented below:

$$
u_{13*5} = \begin{bmatrix}
0 & 0 & 0 & 0 & 1 \\
0 & 0 & 1 & 0 & 0 \\
0 & 0 & 0 & 0 & 1 \\
0 & 0 & 0.275 & 0.725 & 0 \\
0 & 0 & 0.3 & 0.7 & 0 \\
1 & 0 & 0 & 0 & 0 \\
0 & 0 & 0 & 0 & 1 \\
0 & 0 & 0 & 0 & 1 \\
0 & 0 & 0 & 0 & 1 \\
1 & 0 & 0 & 0 & 0 \\
0 & 0.7 & 0.3 & 0 & 0 \\
0 & 0.478 & 0.522 & 0 & 0 \\
0.81 & 0.19 & 0 & 0 & 0
\end{bmatrix}
\tag{15}
$$

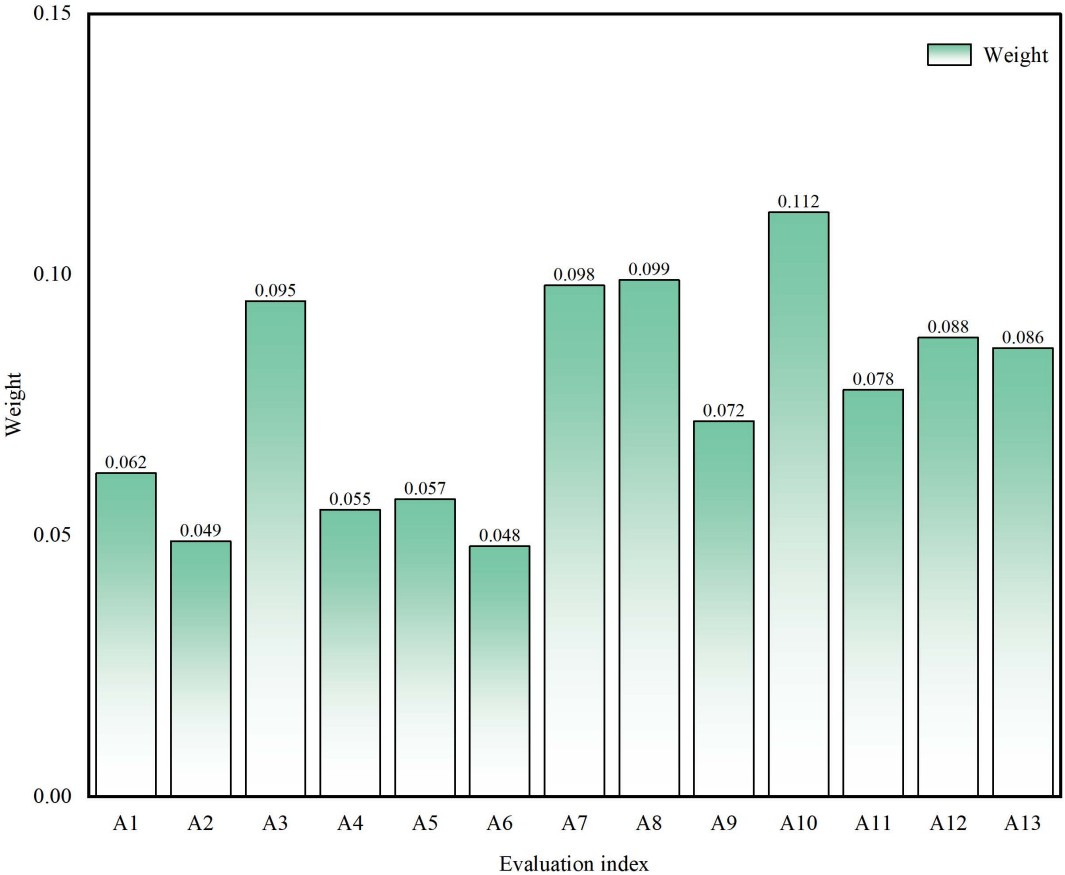

**Fig 2. Weights of indexes at all levels.** The figure presents the magnitude and relative importance of the weights assigned to each indicator, clearly revealing their respective contributions to the overall assessment.

(2) Relative membership degree: Based on Equation 10 and the single-index measurement matrix previously calculated, the relative membership degree of ice flood disaster risk for Jungar Banner is ascertained. The distribution of the relative membership degrees across different levels is illustrated in Fig 4.

(3) Evaluation level determination: Based on Equations 11 and 12 along with the relative membership degree vector of ice flood disaster risk for Jungar Banner previously calculated, yields the level eigenvalue for Jungar Banner's ice flood disaster risk. In the same manner, the level eigenvalues for Tokto County and Qingshuihe County are computed. The results of level eigenvalues under diverse parameter combinations are presented in Fig 5. For Tokto County, the ice flood disaster risk level eigenvalue $\overline{H}$ =2.088, indicating a " medium risk "; For Jungar Banner, the level eigenvalue $\overline{H}$ =3.545, indicating a "high risk"; For Qingshuihe County, the level eigenvalue $\overline{H}$ =2.625, indicating a "medium risk".

## Evaluation results and analysis

According to the evaluation results:

(1) Jungar Banner has the highest risk level of ice flood disaster, which is classified as "high risk". The river in Jungar Banner has a great length, a high degree of curvature, and numerous bridges across it. These characteristics make it

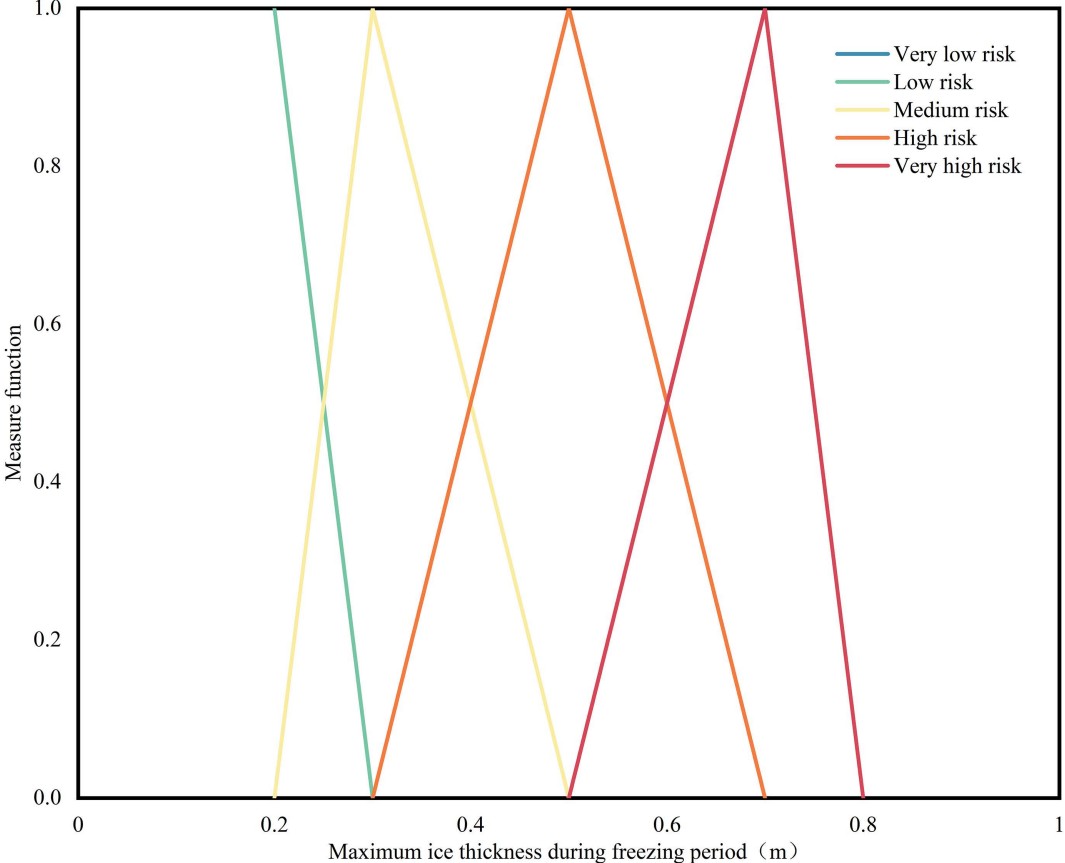

**Fig 3. Measure function of maximum ice thickness during freezing period.** The figure depicts the measure function for maximum ice thickness during the freezing period, shown here as a representative example.

easy for ice to cause waterway obstruction, thereby triggering flood disasters. Hence, the risk of ice flood disaster in this area is considerably higher than that in other areas. According to the historical disaster records, the ice jam bursting in Zhungeer Banner caused flooding in nearby villages and inundated residential houses, which corresponds to the assessment results.

(2) Tokto County has the lowest risk level of ice flood disaster, which is classified as "medium risk". Although Tokto County has the highest population density among the three study areas, it features thinner ice, a lower water level rise, less river curvature, and longer levees. These factors effectively reduce the likelihood and risk of ice flood disasters, resulting in a significantly lower risk value for Tokto County compared to other areas.

(3) Qingshuihe County, classified as a medium-risk area, has a risk value between the other two. In Qingshuihe County, the ice layer is thicker and the water level rises significantly during the freezing period, resulting in a higher flood risk than in Tokto County. However, due to the short river reach in this area, the river has a small curvature and is close to the downstream Wanjiazhai Reservoir, the flood risk can be promptly alleviated by regulating the reservoir's flow. Consequently, the risk value of Qingshuihe County lies between those of the other two regions.

(4) The results were compared with those from the improved catastrophe evaluation method reported in [12]. The latter classified Tokto County as a moderate-risk area (0.644), Jungar Banner as a high-risk area (0.743), and Qingshuihe

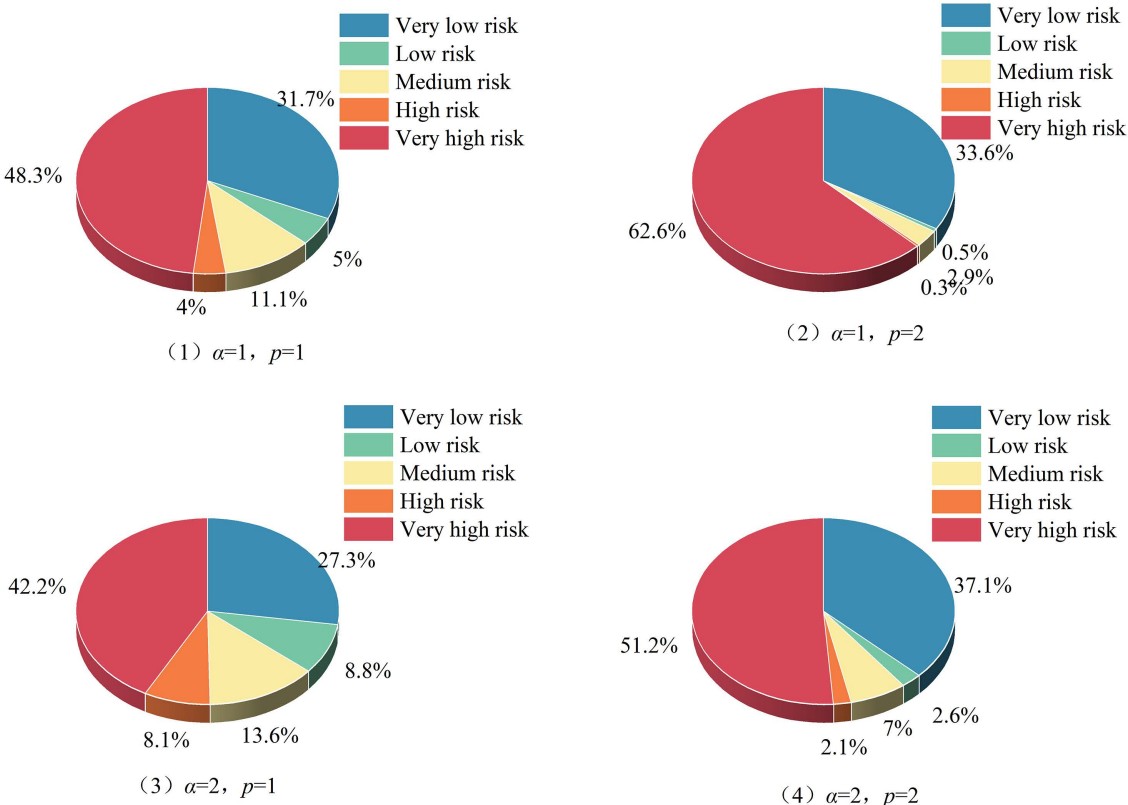

**Fig 4. Relative membership degree of all levels of risk of ice flood disaster in Jungar Banner.** Taking Jungar Banner as an example, the figure compares the comprehensive relative membership degrees obtained under the four alternative parameter combinations.

County as a moderate-risk area (0.676). Both methods produced consistent risk classifications and rankings across the three regions, and the results closely correspond to the actual ice flood conditions in each area. This comparison confirms the reliability of the evaluation method proposed in this study.

## Conclusion

(1)  Based on the analysis of the disaster-causing factors and disaster formation mechanisms of ice flood hazards in the Yellow River Inner Mongolia section, and with reference to existing relevant studies, a risk assessment index system for ice flood disasters was constructed with the regional disaster system theory. This index system effectively reflects the risk characteristics of ice flood disasters in the Yellow River Inner Mongolia section, providing a theoretical basis for regional risk assessments.

(2) Based on the UMT and the VFST, the ice flood risk assessment model for the Yellow River segment in Inner Mongolia was established, and the risk assessment was carried out in 3 typical areas of the Inner Mongolia section. The results showed the following: the ice flood disaster risk level in Tokto County, with a characteristic value of $\overline{H}$ =2.088, was classified as "medium risk"; that in Jungar Banner, with a characteristic value of $\overline{H}$ =3.545, was classified as "high risk"; and that in Qingshuihe County, with a characteristic value of $\overline{H}$ =2.625, was classified as "medium risk".

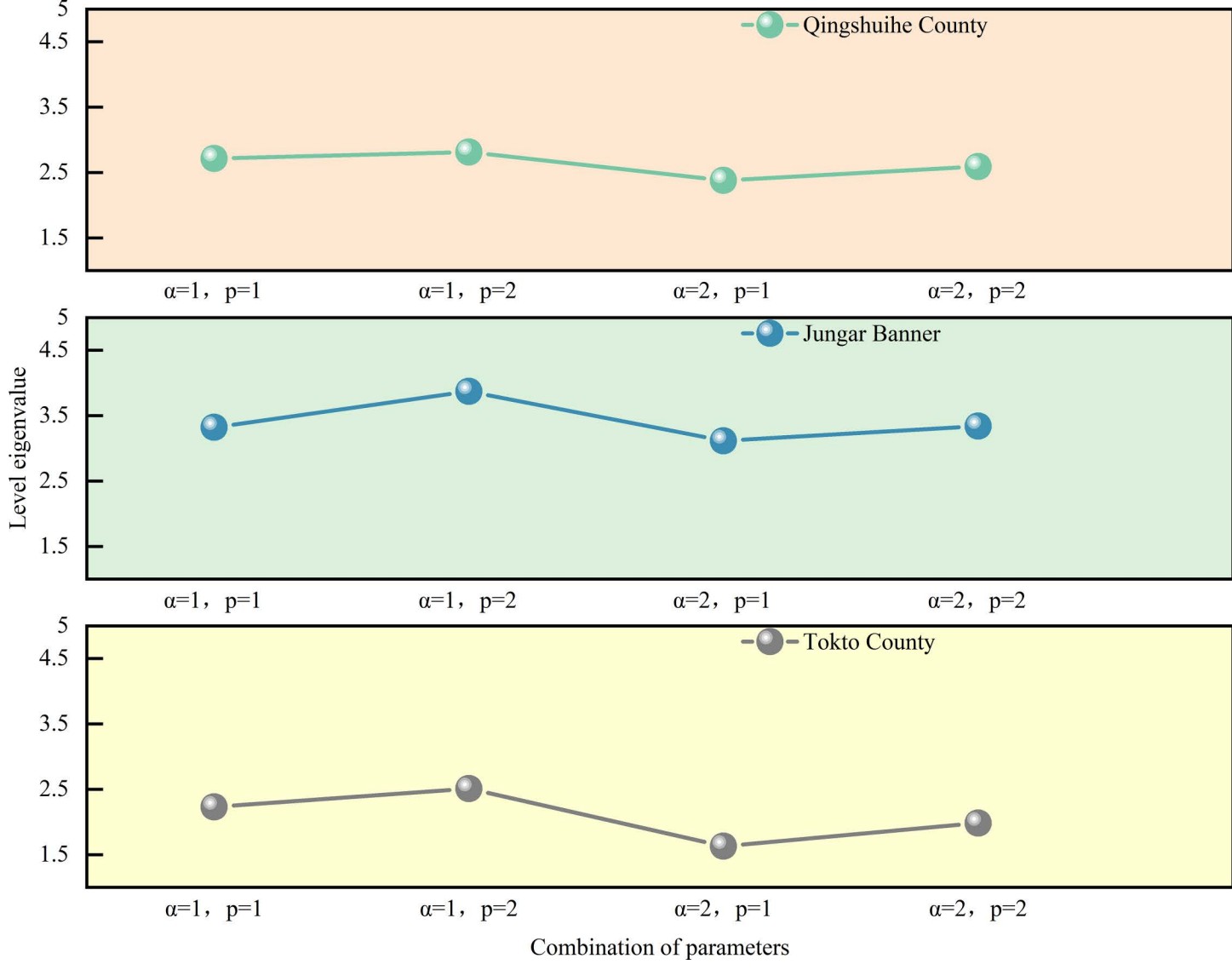

**Fig 5. Level characteristic values under different parameter combinations in each banner county.** The figure displays the level-eigenvalue profiles calculated for the three study areas under each parameter combination.

(3) By utilizing the established risk assessment model, the ice flood disaster risks in the three counties were analyzed and then compared with the results from the improved catastrophe evaluation method. The evaluation results from both methods were consistent regarding the risk rankings, confirming the reliability of the findings. This study offers scientific evidence and effective technical support for risk assessment and flood prevention in the Yellow River Inner Mongolia section, thereby contributing to disaster mitigation efforts.

## Author contributions

**Conceptualization:** Yu deng, Juan Wang.

**Data curation:** Lu Jiang.

**Formal analysis:** Lu Jiang.

**Methodology:** Yu deng, Lu Jiang.

**Writing – original draft:** Yu deng, Lu Jiang.

**Writing – review & editing:** Juan Wang.

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
