## [Decision Letter · Decision Letter 0]

29 Oct 2025

PONE-D-25-44421Ice Flood Disaster Risk Assessment of the Inner Mongolia Reach in the Yellow River Based on Unascertained Measure Theory-Variable Fuzzy SetsPLOS ONE

Dear Dr. deng,

Thank you for submitting your manuscript to PLOS ONE. After careful consideration, we feel that it has merit but does not fully meet PLOS ONE’s publication criteria as it currently stands. Therefore, we invite you to submit a revised version of the manuscript that addresses the points raised during the review process.

We look forward to receiving your revised manuscript.

Kind regards,

Sanjay Kumar

Academic Editor

PLOS ONE

Journal Requirements:

“This work was supported by the National Natural Science Foundation of China under grant U23A2012, the Yellow River Water Conservancy Research Institute basic research funds under grant HKY-JBYW-2022-08.”

5. We note that your Data Availability Statement is currently as follows: All relevant data are within the manuscript and in Supporting Information files.

6. We note that Figure 1 in your submission contain map/satellite images which may be copyrighted. All PLOS content is published under the Creative Commons Attribution License (CC BY 4.0), which means that the manuscript, images, and Supporting Information files will be freely available online, and any third party is permitted to access, download, copy, distribute, and use these materials in any way, even commercially, with proper attribution. For these reasons, we cannot publish previously copyrighted maps or satellite images created using proprietary data, such as Google software (Google Maps, Street View, and Earth). For more information, see our copyright guidelines: http://journals.plos.org/plosone/s/licenses-and-copyright.

Reviewers' comments:

Reviewer's Responses to Questions

**Comments to the Author**

1. Is the manuscript technically sound, and do the data support the conclusions?

Reviewer #1: Yes

Reviewer #2: Yes

Reviewer #3: Yes

2. Has the statistical analysis been performed appropriately and rigorously? 

Reviewer #1: Yes

Reviewer #2: Yes

Reviewer #3: Yes

3. Have the authors made all data underlying the findings in their manuscript fully available?

Reviewer #1: Yes

Reviewer #2: Yes

Reviewer #3: Yes

4. Is the manuscript presented in an intelligible fashion and written in standard English?

Reviewer #1: Yes

Reviewer #2: Yes

Reviewer #3: Yes

5. Review Comments to the Author

Reviewer #1: Overall, I like the article which I consider an interesting contribution to Disaster Risk Assessment.

I do have some minor comments

The introduction section has too long paragraphs, please separate them into shorter paragraphs for the good sake of the reader

On lines 116-120 authors explain the three-tiered evaluation system, but they use different expressions on those lines and the figure, for instance in line 118-119 they list: disaster-inducing factors, disaster-predisposing conditions, hazard-bearing entities, and disaster mitigation capacity. However, in Figure 2 they list: disaster-inducing factors; disaster-breeding environment, hazard bearing bodies, and disaster response capability. That difference leads to confusion and must be emended

Line 144, 161, 225, I do not understand the symbology, please check the edition of the pdf document.

Reviewer #2: This paper presents an innovative approach by integrating Uncertainty Measure Theory (UMT) with Variable Fuzzy Set Theory (VFST) to develop a model for assessing ice flood risk in the Inner Mongolia reach of the Yellow River. The study shows strong originality, but some revisions are needed:

1. In UMT-VFST Evaluation Method, the division of risk into five levels lacks a clear reference to specific standards. Please provide the source of the classification (e.g., national standards, industry guidelines, or authoritative literature).

2. The terminology for ice floods is inconsistent (e.g., ice flood vs. ice-jam flood). Consistency in terminology should be ensured throughout.

3. Issues in the tables: The header row uses indexes instead of the singular index, and the overall formatting (column width, alignment, borders) reduces readability. Please unify the use of index and improve formatting to meet academic publishing standards.

4. There is inconsistent use of Chinese and English punctuation throughout the manuscript. For example, in line 231, the Chinese full-width parenthesis “（” is used instead of the half-width “(”. It is recommended to unify the use of half-width punctuation marks, and carefully check all parentheses, commas, periods, etc. throughout the text.

Reviewer #3: This paper couples Uncertainty Measure Theory with Variable Fuzzy Set Theory to address the complex characteristics of disaster systems, thereby enabling a more accurate analysis and assessment of ice flood disasters along the Inner Mongolia reach of the Yellow River. The study demonstrates strong originality, but certain aspects still require improvement.

1 In Section 'Weight calculation', the exclusive reliance on the Analytic Hierarchy Process (AHP) for weight determination raises concerns regarding methodological robustness. Given that AHP is inherently subjective due to its dependence on expert judgment, it is recommended to incorporate objective weighting methods (e.g., entropy weight method, CRITIC) to enhance result reliability.

2 In Figure 3, the font sizes are inconsistent—for example, the axis labels, legend, and data annotations use different sizes, which reduces readability. It is recommended to check all figures in the manuscript and ensure uniform font style and size.

3 The use of thousand separators in the data is inconsistent (e.g., in Table 1). Please review and correct this throughout the manuscript.

4 All variables, parameters, and function names should be formatted consistently in either italics or roman type, in accordance with academic conventions. Particular attention should be paid to font usage in the Method section.

5 In the Weight Calculation section, paragraph formatting is inconsistent. Please revise according to the journal’s formatting guidelines.

6. PLOS authors have the option to publish the peer review history of their article (what does this mean?). If published, this will include your full peer review and any attached files.

Reviewer #1: **Yes:** Jose Luis Arumi

Reviewer #2: No

Reviewer #3: No

---

## [Author Response · Author response to Decision Letter 1]

10 Dec 2025

Reviewer #1: Overall, I like the article which I consider an interesting contribution to Disaster Risk Assessment.I do have some minor comments

1.The introduction section has too long paragraphs, please separate them into shorter paragraphs for the good sake of the reader

Answer: Thank you for your valuable comment. As you pointed out, the introduction in our previous submission contained overly long paragraphs, which may have hindered readability. We have now reorganized and divided the introduction into shorter, more coherent paragraphs to improve clarity and enhance the overall reading experience.

2.On lines 116-120 authors explain the three-tiered evaluation system, but they use different expressions on those lines and the figure, for instance in line 118-119 they list: disasterinducing factors, disaster-predisposing conditions, hazard-bearing entities, and disaster mitigation capacity. However, in Figure 2 they list: disaster-inducing factors; disaster-breeding environment, hazard bearing bodies, and disaster response capability. That difference leads to confusion and must be emended

Answer: Thank you for your professional question. We have carefully revised the manuscript to address the inconsistencies in terminology. The expressions disaster-inducing factors, disaster-breeding environment, hazards-bearing body, and disaster response capability are now used consistently throughout the paper.

3.Line 144, 161, 225, I do not understand the symbology, please check the edition of the pdf document.

Answer: Thank you for your professional question. In the previous version, we did not provide sufficiently clear explanations for some of the symbols introduced in the manuscript, which may have affected readers’ understanding. We have now added detailed explanations and clarifications to improve the readability of this section.

Reviewer #2: This paper presents an innovative approach by integrating Uncertainty Measure Theory (UMT) with Variable Fuzzy Set Theory (VFST) to develop a model for assessing ice flood risk in the Inner Mongolia reach of the Yellow River. The study shows strong originality, but some revisions are needed:

1. In UMT-VFST Evaluation Method, the division of risk into five levels lacks a clear reference to specific standards. Please provide the source of the classification (e.g., national standards,industry guidelines, or authoritative literature).

Answer: Thank you for your professional comments! At present, there are no clear norms or unified classification standards for ice flood disaster risk assessment. Therefore, this study refers to the mature classification methods in the field of flood disaster risk assessment, divides the ice flood disaster risk levels into five grades, and has supplemented relevant explanations in the manuscript.

2. The terminology for ice floods is inconsistent (e.g., ice flood vs. ice-jam flood). Consistency in terminology should be ensured throughout.

Answer: Thank you for your professional insight. At present, there is no standardized or universally accepted classification system for ice flood disaster risk assessment. Therefore, in this study, we refer to the classification methods commonly used in flood disaster risk assessment and divide ice flood disaster risk into five levels. We have added corresponding explanations in the revised manuscript.

3. Issues in the tables: The header row uses indexes instead of the singular index, and the overall formatting (column width, alignment, borders) reduces readability. Please unify the use of index and improve formatting to meet academic publishing standards.

Answer: Thank you for your professional comment. We have carefully reviewed the tables and corrected all non-standard elements accordingly.

4. There is inconsistent use of Chinese and English punctuation throughout the manuscript. For example, in line 231, the Chinese full-width parenthesis “（” is used instead of the halfwidth “(”. It is recommended to unify the use of half-width punctuation marks, and carefully check all parentheses, commas, periods, etc. throughout the text.

Answer: Thank you for your professional comment. We have thoroughly reviewed the formatting and layout of the manuscript and revised them in accordance with the journal’s formatting requirements.

Reviewer #3: This paper couples Uncertainty Measure Theory with Variable Fuzzy Set Theory to address the complex characteristics of disaster systems, thereby enabling a more accurate analysis and assessment of ice flood disasters along the Inner Mongolia reach of the Yellow River. The study demonstrates strong originality, but certain aspects still require improvement.

1. In Section 'Weight calculation', the exclusive reliance on the Analytic Hierarchy Process (AHP) for weight determination raises concerns regarding methodological robustness. Given that AHP is inherently subjective due to its dependence on expert judgment, it is recommended to incorporate objective weighting methods (e.g., entropy weight method, CRITIC) to enhance result reliability.

Answer: Thank you for your professional comment. As you pointed out, relying solely on subjective weighting methods may affect the accuracy of the evaluation results. Therefore, following your suggestion, we incorporated the entropy weight method as an objective weighting approach. By combining both subjective and objective weights, we improved the reliability of the final assessment results.

2. In Figure 3, the font sizes are inconsistent—for example, the axis labels, legend, and data annotations use different sizes, which reduces readability. It is recommended to check all figures in the manuscript and ensure uniform font style and size.

Answer: Thank you for your professional comment. We have revised the non-standard formatting issues in the figures and tables to ensure full compliance with the journal’s requirements.

3. The use of thousand separators in the data is inconsistent (e.g., in Table 1). Please review and correct this throughout the manuscript.

Answer: Thank you for your professional comment. We have carefully reviewed all data formats in the manuscript and corrected any irregularities.

4. All variables, parameters, and function names should be formatted consistently in either italics or roman type, in accordance with academic conventions. Particular attention should be paid to font usage in the Method section.

Answer: Thank you for your professional comment. The formatting of all variables, parameters, and function names in the manuscript has been revised accordingly.

5. In the Weight Calculation section, paragraph formatting is inconsistent. Please revise according to the journal’s formatting guidelines.

Answer: Thank you for your professional comment. The formatting of the Weight caught section has been revised accordingly.

We would like to express our sincere gratitude to all editors and experts for their careful guidance! We have strictly revised the manuscript in accordance with the review comments. Should there be any aspects that still need improvement, we sincerely request further corrections and look forward to your reply.

---

## [Decision Letter · Decision Letter 1]

13 May 2026

Ice Flood Disaster Risk Assessment of the Inner Mongolia Reach in the Yellow River Based on Unascertained Measure Theory-Variable Fuzzy Sets

PONE-D-25-44421R1

Dear Dr. deng,

We’re pleased to inform you that your manuscript has been judged scientifically suitable for publication and will be formally accepted for publication once it meets all outstanding technical requirements.

Kind regards,

Beata Calka, PH.D.

Academic Editor

PLOS One

Additional Editor Comments (optional):

Before final acceptance, editorial revisions are required, particularly in Table 1, to ensure full consistency and uniformity in the presentation of data throughout the manuscript.

Reviewers' comments:

Reviewer's Responses to Questions

**Comments to the Author**

1. If the authors have adequately addressed your comments raised in a previous round of review and you feel that this manuscript is now acceptable for publication, you may indicate that here to bypass the “Comments to the Author” section, enter your conflict of interest statement in the “Confidential to Editor” section, and submit your "Accept" recommendation.

Reviewer #1: All comments have been addressed

Reviewer #3: All comments have been addressed

2. Is the manuscript technically sound, and do the data support the conclusions?

Reviewer #1: Yes

Reviewer #3: Yes

3. Has the statistical analysis been performed appropriately and rigorously? 

Reviewer #1: Yes

Reviewer #3: Yes

4. Have the authors made all data underlying the findings in their manuscript fully available?

Reviewer #1: Yes

Reviewer #3: Yes

5. Is the manuscript presented in an intelligible fashion and written in standard English?

Reviewer #1: Yes

Reviewer #3: Yes

6. Review Comments to the Author

Reviewer #1: All my previous comments were satisfactory addressed and the article is sound. Therefore, I have no further comments.

Reviewer #3: The author has provided reasonable explanations and adequate responses to the five comments raised in the first review. Similarly, the author has made serious, systematic, and comprehensive improvements based on the comments from the other two reviewers. It is suggested that some details of the manuscript be further refined. For instance, in Table 1 the numerical ranges expressed in two forms—basic mathematical notation and statistical/quality management notation. It is suggested that the author use a consistent format for expressing numerical ranges in Table 1.

7. PLOS authors have the option to publish the peer review history of their article (what does this mean?). If published, this will include your full peer review and any attached files.

Reviewer #1: No

Reviewer #3: No

---

## [Editor Report · Acceptance letter]

PONE-D-25-44421R1

PLOS One

Dear Dr. deng,

I'm pleased to inform you that your manuscript has been deemed suitable for publication in PLOS One. Congratulations! Your manuscript is now being handed over to our production team.

Kind regards,

on behalf of

Dr. Beata Calka

Academic Editor

PLOS One